# Fatigue Cracking Evolution and Model of Cold Recycled Asphalt Mixtures during Different Curing Times

**DOI:** 10.3390/ma15134476

**Published:** 2022-06-24

**Authors:** Yu Xia, Juntao Lin, Zongwu Chen, Jun Cai, Jinxiang Hong, Xiaobin Zhu

**Affiliations:** 1Faculty of Engineering, China University of Geosciences (Wuhan), Wuhan 430074, China; xiayu874696247@163.com (Y.X.); chenzw@cug.edu.cn (Z.C.); 2Key Laboratory of Road Structure and Material of Ministry of Transport (Changsha), Changsha University of Science & Technology, Changsha 410205, China; juncai0@163.com; 3State Key Laboratory of High Performance Civil Engineering Materials, Jiangsu Research Institute of Building Science, Nanjing 210008, China; hongjinxiang@cnjsjk.cn (J.H.); xbzhu@163.com (X.Z.)

**Keywords:** cold recycled mixtures, fatigue cracking, SCB test, fatigue damage model

## Abstract

This paper aims to investigate the fatigue cracking evolution of cold recycled asphalt mixtures with asphalt emulsion (CRME) under different curing times. The fatigue cracking model of CRME based on damage mechanics and fracture mechanics was analyzed according to the fatigue loading curve. Firstly, the fatigue cracking evolution of CRME was studied through an SCB strength test and SCB fatigue test. Then, the fatigue damage mechanics were used to establish a nonlinear fatigue cracking model, and the damage degree of CRME at the initial cracking point was determined. The Paris formula was used to characterize the law of fatigue crack propagation. Finally, the microstructure of CRME was observed by scanning electron microscopy (SEM) with the backscattering method. The results indicate that the initial cracking point appears at around 60% of the fatigue life according to the SCB fatigue test by means of image analysis. The damage variable was obtained through the cracking model, and the value of the damage variable was determined as 0.06–0.17 at the initial cracking point. In addition, the Paris formula showed that the crack growth of CRME can be reflected by the stress intensity factor and correlative parameters. Moreover, cement hydration products were mixed with the asphalt membrane to form a denser spatial structure during the curing process, which may provide higher fatigue performance of CRME. This research may provide a theoretical reference for studying the fatigue cracking behavior of CRME.

## 1. Introduction

A large amount of reclaimed asphalt pavement (RAP) is produced every year, which not only takes up a lot of land but also causes serious pollution to the environment. Therefore, cold recycled technology has been rapidly developed all over the world due to its advantages in resources and environments, such as the high RAP utilization ratio, energy conservation, and emissions reduction [1,2]. However, the cold recycled mixtures may have poor fatigue cracking resistance due to their high air void (8–14%) [3]. On the other hand, cold recycled mixtures are gradually used in expressways with high traffic loading in China, and thus its fatigue cracking resistance deserves more attention at present. 

Many researchers have focused on the influencing factors of cold recycled mixtures and performed fatigue tests based on laboratory methods. Leandri et al. [4] researched the effect of cement on the fatigue cracking resistance of CRME and found that cement could improve the fatigue cracking resistance under low strain levels and reduce that under high strain levels. Yan et al. [5] researched the fatigue performance of CRME through indirect tensile fatigue tests and drew the same conclusion. Lin et al. [6] studied the dynamic characteristics of CRME through microscopic and mesoscopic perspectives and found that the fatigue life of four-point bending could be influenced significantly by the curing time (7 d, 28 d at 20 °C). Wang et al. [7] proposed that both the penetration of asphalt emulsion and the RAP content had a significant impact on the fatigue life of CRME. Zhao et al. [8] analyzed the fatigue characteristics of CRME by considering the composition of recycled aggregate. It was found that the fatigue life of CRME decreased as the recycled aggregate content and stress levels increased.

Some research was performed on the fatigue cracking evolution and its models of CRME. Gao et al. [9] researched the propagation of fatigue cracks of CRME by means of the digital image method. The two turning points in the fatigue curve are verified as the initial cracking point and fatigue failure point, according to the results. Jiang et al. [10] investigated the fatigue properties of CRME based on indirect tensile fatigue tests. The fatigue model, established on the basis of the Weibull distribution, could effectively assess the fatigue life of CRME. Chelelgo et al. [11] considered the combined effects of temperature and time on the fatigue-strength development of CRME through four-point bending fatigue tests. A parabolic hyperbolic maturity model was established to predict the fatigue strength of CRME. Kavussi et al. [12] developed a fatigue model for CRME based on extensive indirect tensile fatigue and resilient modulus tests. It is also indicated that the fatigue life increased at low strain levels with the cement content increased.

Based on the previous studies, researchers have comprehensively investigated the influencing factors and fatigue models of CRME. However, the analysis of the fatigue behavior and existing models for CRME have not considered the combined effect of damage mechanics and fracture mechanics. Besides, there is less attention on the fatigue cracking evolution and fatigue models of CRME, which has provided the motivation to undertake this work.

The objective of this paper is to investigate the fatigue cracking evolution of cold recycled mixtures under different curing times. The models of fatigue cracking evolution with different curing times are based on image analysis and the fatigue loading curve. The damage mechanics theory is used to establish the fatigue evolution model before the occurrence of initial fatigue cracks, while the propagation behavior of fatigue cracks from the initial cracking point to the fatigue failure point is analyzed based on fracture mechanics. 

## 2. Materials and Experiments

### 2.1. Materials

#### 2.1.1. RAP Materials

The quality and content of RAP material have a great influence on the mechanical properties and road performance of CRME. According to the standard test methods of bituminous mixtures for highway engineering (JTG E20-2011) [13], the HMA with a nominal maximum size of 19 mm was designed and prepared in a laboratory, and then the loose asphalt mixtures were aged in an oven at 135 °C for 18 h to simulate the long-term aging. The prepared RAP material was divided into three groups (0–5 mm, 5–10 mm, and 10–20 mm), and the RAP aggregates gradation of CRME is listed in Table 1.

#### 2.1.2. Asphalt Emulsion

One kind of emulsifier produced by Jiangsu province was selected to prepare the cationic slow-setting (CSS) asphalt emulsion in the laboratory with a colloid mill. The properties of asphalt emulsion are listed in Table 2. The droplet size distribution of asphalt emulsion is shown in Figure 1.

#### 2.1.3. The Gradation of Mixtures

Conventional Portland cement (PO.42.5) was selected from the market. As an important active filler in CRME, cement can significantly increase the early strength and moisture damage resistance of CRME. The content of cement in CRME was designed as 1.5% by weight of aggregates. The gradation of CRME with 82% RAP was designed by modified Marshall methods according to the specification of China (JTG/T 5521-2019) [14]. Some new aggregates were added in order to improve the gradation of CRME, and the gradation of CRME is shown in Figure 2. Then, the optimum adding moisture content was determined as 1.6% by the maximum dry density method, and the optimum asphalt emulsion content was determined as 4.0% based on the mechanical tests. Table 3 lists the composition of aggregate in CRME.

### 2.2. Material Preparation

According to the design results of CRME, cylindrical samples were prepared with an SGC (Superpave gyratory compactor) by 50 compaction times with a mold of 150 mm in diameter, and the process was the same as the previous study [15]. The weight of each cylindrical sample was about 3820 g, and the height was controlled to 100 ± 3 mm. In order to better simulate the curing condition, the plastic film was wrapped on the bottom and sides of the cylinder after the samples were prepared. In this way, the moisture inside the samples evaporated only through the top surface. Then, the samples were placed in a curing room to maintain a temperature of (20 ± 2) °C and a humidity for 7-days, 14-days, 28-days, and long-term curing until testing. It was noticed that the samples with a long-term curing time were simulated by aging in an oven at 60 °C for 3 d [16]. When reaching the curing time, each cylindrical sample was cut into four semicircular samples with a thickness of 50 mm. Then, a pre-cut seam in the middle of the semicircular sample was cut. Considering the experiment conditions and the previous research in SCB tests [17,18,19], the size of the prefabricated slit was 2 mm in width and 15 mm in depth. The samples were painted white to better capture cracking behavior during the tests. The preparation of the samples is shown in Figure 3.

### 2.3. Experiments

#### 2.3.1. SCB Strength Test

The SCB test was used to obtain the strength and characterize the fracture resistance of CRME. A universal testing machine (UTM-100), equipped with a temperature control chamber, was used for the SCB test. The SCB test loading system and the force mode are shown in Figure 4. The loading device is mainly composed of a loading roller at the midpoint of the semi-circular arch and two supporting rollers at the flat edge. It was noted that the span over the two supporting rollers was 120 mm, which was 0.8 times the specimen’s diameter. The tests were carried out at 15 °C with a loading rate of 50 mm/min, and two replicates of the samples were tested for each curing time. According to EN 12697-44 [20], the maximum stress at failure $\sigma_{\max}$ was calculated in accordance with Equation (1): (1)$$\sigma_{\max}=\frac{4.263\times F_{\max}}{D\times t}$$
where *F*_max_ is the maximum force, *N*; *D* is the diameter of the sample, mm; and *t* is the thickness of the sample, mm.

#### 2.3.2. SCB Fatigue Test

The SCB fatigue test was also conducted by UTM-100. The haversine wave load, with a loading frequency of 2 Hz and test temperature of 15 °C, was chosen during the fatigue test. The crack evolution process was recorded under stress control conditions during the test, and different stress levels (0.2, 0.3, 0.4, 0.5, 0.6) were chosen [17]. A CMOS industrial camera was used to record the full process of the SCB fatigue test. In addition, in order to better capture the fatigue cracking point of CRME during the fatigue test, a sampling frequency of 4 Hz was set, and 2 images were collected in each fatigue loading cycle.

## 3. Result and Discussion

### 3.1. SCB Strength

The load-displacement curve of CRME at different curing times by the SCB test is shown in Figure 5. As presented, the fracture strength of CRME significantly increased with the increase of the curing time. The CRME with accelerated curing has a significantly higher strength, while the CRME with 7-day curing has the lowest strength. Moreover, it is noted that the curve of the CRME with 7-day curing is different from other samples, and the drop-down curve after reaching maximum strength is not obvious. The results indicate that the CRME with 7-day curing has not formed enough strength, and the inner structure is similar to granular materials. The results of the SCB strength are consistent with the indirect tensile strength (ITS) test in the previous study [17]. In addition, it is well known that the strength of CRME has continuously developed with the increase of the curing time. The results show that the fracture strength of CRME after 28-days of curing is slightly lower than that of CRME with accelerated long-term curing, which indicates that the strength performance of CRME with a 28-day curing approach to that of long-term curing. 

The fracture energy of the samples can be calculated according to the load-displacement curve, which is listed in Formula (2). Based on the previous studies [21,22], the crack resistance of asphalt mixtures can be evaluated by the flexibility index (FI) and the crack resistance index (CRI). The FI and CRI are shown in Formulas (3) and (4). It is worth noting that both the FI and CRI are derived from fracture energy.
(2)$$G_{f}=\frac{W_{f}}{A}=\frac{\int Fdu}{A}$$
(3)$$FI=\frac{G_{f}}{\left|m\right|}\times 0.01$$
(4)$$CRI=\frac{G_{f}}{F_{\max}}$$
where *G_f_* is the fracture energy, J/m^2^; *W_f_* is the work done by the load, representing the area of the region enclosed by the load-displacement curve and the displacement axis, *J*; *A* is the projected area of the fracture surface, A=(W−a)×t, where *W* is the height of the semicircular sample(radius), *a* is the length of the prefabricated slit, *t* is the thickness of the sample; *FI* is the Illinois index, dimensionless; *m* is the post-peak slope, kN/mm; *CRI* is the crack resistance index, m^−1^.

Compared with fracture energy, the fracture toughness has a simpler form, which is calculated based on the peak load and the geometric shape factor of the samples, as shown in Formulas (5)–(7).
(5)$$\sigma_{\max}=\frac{4.8\times F_{\max}}{D\times t}$$
(6)$$K_{IC}=\sigma_{\max}\sqrt{\pi a}f\left(\frac{a}{W}\right)$$
(7)$$f\left(\frac{a}{W}\right)=4.782+1.219\left(\frac{a}{W}\right)+0.063^{7.045\left(\frac{a}{W}\right)}$$
where $\sigma_{\max}$ is the maximum fracture tensile stress, MPa; *F*_max_ is the peak load, *N*; *D* is the diameter of the SCB sample, mm; *T* is the thickness of the sample, mm; $K_{IC}$ is the fracture toughness of the I-mode crack, MPa·mm^1/2^; f(a/W) is the geometric factor of the sample [23,24], dimensionless; a is the depth of the pre-cut, mm; *W* is the height of the SCB sample, mm.

According to the SCB test results of the CRME with different curing times, the parameters that characterize its fracture resistance are calculated in Table 4. As shown in Table 4, the cold recycled mixtures at the 7-day curing time show a very low strength, and the strength gradually increases with the increase of the curing time. However, the increase rate is gradually reduced. It is found that when *G_f_*, *FI*, and CRI are used to characterize the cracking resistance performance of CRME, the sample with the 7-day curing is better than that of the 14-day due to the properties of loose material. In addition, the *G_f_*, *FI*, and *CRI* values decrease with the curing time increase in both the 28-day and long-term curing. It is related to the area enclosed by the load-displacement curve and the large difference in the slope of the inflection point after the peak. Therefore, it is more convincing to use fracture toughness to characterize the crack resistance of cold recycled mixtures.

### 3.2. Fatigue Life and Fatigue Equation

The typical fatigue curve of asphalt mixtures is shown in Figure 6. It can be seen that the curve presents a relatively obvious reverse “S” shape with two “turning points.” The turning point is determined by the intersection of two tangents. Generally, the first turning point of the fatigue curve is defined as the initial cracking point, and the second turning point is defined as the fatigue failure point [9,25]. According to these two turning points, the crack propagation behavior is divided into three stages. The first stage is the crack initiation stage, the second stage is the crack growth stage, and the third stage represents the crack accelerated growth stage. The indenter displacement increases sharply with the increase of loading times after the second turning point. The fatigue cracks expand rapidly and finally penetrate the sample; then, the sample completely loses its load-bearing capacity. Therefore, the second turning point is defined as the fatigue failure point, and the number of loading cycles at the fatigue failure point is defined as the fatigue life of the CRME. Table 5 shows the fatigue life of cold recycled mixtures with different curing times under different stress ratios. 

As shown in Table 5, the stress level under the same curing time has a great influence on the fatigue life, especially at the 14-day curing. It can be seen that the fatigue life is longer under the low-stress ratio, while the fatigue life is shorter under the high-stress ratio. The fracture behavior under the high-stress ratio almost shows a brittle fracture, and it penetrates the sample quickly once the crack appears. On the other hand, it is found that the curing time also has a significant effect on the fatigue life under the same ratio. The fatigue life increases with the increase of the curing time. 

In order to depict the fatigue characteristics of CRME at a wider stress level, the fatigue life of CRME at different curing times can be calculated by the following equation:(8)$$N_{f}=k{\left(\frac{1}{\sigma }\right)}^{n}$$
(9)$$\mathrm{lg}N_{f}=\mathrm{lg}k-n\mathrm{lg}\sigma$$
where *N_f_* is the cycle number to failure; *σ* is the test stress; *k* and *n* are the fatigue regression parameters. 

Based on the data in Table 5 and the equations above, the fitting results are listed in Table 6. The value of the regression coefficient “*k*” in the fatigue equation can reflect the line level of the fatigue curve. The line position of the fatigue curve increases with the increase of the *k* value, and the fatigue performance will also increase. According to the fitting results in Table 6, it can be seen that the value of *k* in the fatigue equation increases with the increase of the curing time, which indicates that the fatigue performance gradually improves with the increase of the curing time. 

### 3.3. Fatigue Cracking Point Analysis

The surface crack propagation behavior of the samples during the SCB fatigue test is emphatically analyzed in the paper. The initial cracking point, identified by image analysis, is defined as the point where mesoscopic cracks first appear on the surface of the CRME sample. The second turning point in the fatigue loading curve is defined as the fatigue failure point, and the fatigue failure point is shown in Figure 7.

The behavior of fatigue crack growth is shown in Figure 8 and Figure 9. Figure 8 shows the position of the initial cracking point with different stress ration and different curing times, while the fatigue failure point of the samples is shown in Figure 9. The number of loading cycles is displayed in the top corner of the pictures.

The ratio of the number of loading cycles between the initial cracking point to the fatigue failure point of CRME is shown in Table 7. It can be seen from Table 7 that the ratio with the 14-day curing time is irregular. In contrast, the ratio of the 28-day curing time is relatively close under the different stress ratios. Considering the effect of the number of loading cycles, the range of the ratio is around between 0.4–0.7. In general, the initial fatigue cracking point appears at around 60% of the fatigue life. It also noted that the ratio shows a decreasing trend, with the increase of the curing time under the same stress ratio. It can be concluded that the fatigue cracking resistance of CRME decreases with the increase of the stress level at the same curing time and increases with the increase of the curing time at the same stress level. The results also verify that the CRME is sensitive to the stress level in the early-stage, and it is prone to more serious diseases after initial cracks appear. 

### 3.4. Fatigue Damage Model Based on Damage Mechanics

The internal damage of CRME gradually accumulates and forms the microscopic fatigue cracks under cyclic loading. To further investigate the fatigue damage behavior, damage mechanics are used to research the damage evolution rule of fatigue cracks.

The damage mechanics theory mainly focuses on the appearance and evolution of the fatigue cracks, and it is vital to define the damage variable. Kachanov (1958) believes that the reduction of the effective bearing area due to material defects is the main mechanism of material deterioration. The damage variable is defined as the reduced degree of the effective bearing area of the material and member, as shown in Formula (10): (10)$$D=1-\frac{A^{\sim }}{A}$$
where *D* is the damage variable; $A^{\sim }$ is the effective bearing area in the damaged state; *A* is the initial bearing area, which means the bearing area in the undamaged state.

According to the fatigue damage study of asphalt pavement, the Miner linear fatigue damage model is widely used to define the damage variable, *D*:(11)$$D\left(N\right)=\frac{N}{N_{f}}$$
where *N* is the number of loading cycles; *N_f_* is the fatigue life.

It can be seen from Formula (11) that the Miner linear fatigue damage model considers that the fatigue damage evolves linearly under the same load conditions. The general form of the damage evolution equation is as follows [26]:(12)$$\frac{dD}{dN}=f\left(\sigma \;\;\;or\;\;\;\varepsilon \;\;\;or\;\;\;W,v,\cdots \right)$$
where *D* is the damage variable; *N* is the number of loading cycles; *W* is the dissipated energy; *v* is the variable reflecting the environmental influence factors. 

Chaboche proposed a fatigue damage model based on continuity damage mechanics and the dissipative energy theory of thermodynamics, as shown in Formula (13).
(13)$$\frac{dD}{dN}={\left[1-{\left(1-D\right)}^{1+\beta }\right]}^{\alpha }{\left[\frac{\sigma_{M}-\bar{\sigma }}{M\left(1-D\right)}\right]}^{\beta }$$
where $\bar{\sigma }$ is the average stress; α, β, and *M* are the material parameters related to temperature. 

Formula (13) can be simplified to Formula (14).
(14)$$\frac{dD}{dN}=a{\left[\frac{\sigma }{1-D}\right]}^{n}$$

Equation (15) can be obtained by the integral Equation (14).
(15)$$D\left(N\right)=1-{\left[1-a(1+p)\sigma^{p}N\right]}^{\frac{1}{1+p}}$$

Equation (16) can be obtained when *D* (*N*) = 1 and *N* = *N_f_*.
(16)$$\frac{1}{1+p}=a\sigma^{p}N_{f}$$

Equation (17) can be obtained by substituting Equation (16) into Equation (15).
(17)$$D\left(N\right)=1-{\left[1-\frac{N}{N_{f}}\right]}^{\frac{1}{1+p}}$$

Then, let *p* = *n*, $k=\frac{1}{a\left(1+p\right)}$, and Equation (18) can be obtained by transforming Equation (16).
(18)$$N_{f}=k{\left(\frac{1}{\sigma }\right)}^{n}$$

Generally, the parameter *n* of Equation (18) is determined by fitting the fatigue equation with SN data. In this paper, the parameter n, with the 14-day, 28-day, and long-term curing time, is determined in Table 6. Equation (17) is obtained by substituting the parameter *n* and *N_f_*, and the fatigue damage model based on damage mechanics are shown in Figure 10. It can be seen that the damage variables increase sharply as the increase of loading cycles are under the high-stress ratio at the same curing time. While at the same stress ratio, the curve of the damage variables shows a faster increase as the curing time decreases. It means the CRME has a lower damage degree at low-stress levels and long curing times. Ignoring some small number of loading cycles, then the ratio of the “initial cracking point/fatigue life” (*N/N_f_*) between 0.38–0.73 is taken and substituted in the damage evolution equations. The damage variables are obtained and shown in Table 8. It can be seen that the damage variable is only between 0.06 to 0.17 when the initial cracking point appears. It shows that even though the number of loading cycles reached 70% of the fatigue life, the damage degree of CRME is still very low. In general, the position of the initial cracking point has a great influence on the fatigue performance of CRME, and thus it deserves more attention in future studies.

### 3.5. Fatigue Damage Model Based on Fracture Mechanics

The fatigue damage of the sample is still evolving, and the fatigue cracks are still expanding after the initial crack generation. Damage mechanics are not suitable to depict the expansion of the fatigue cracks for CRME after the initial crack generation. The fracture mechanics theory is used to study the propagation behavior of fatigue cracks after obvious cracks appear. According to fracture mechanics [27], the ratio of “Δa/ΔN” in one stress cycle can be defined as the fatigue crack growth rate expressed in differential d*a*/d*N* under extreme conditions. It is noticed that the crack length “*a*” means the length perpendicular to the stress direction instead of the actual length of the fatigue crack. Under uniaxial cyclic alternating stress, the crack growth rate perpendicular to the stress direction can generally be written as follows:(19)$$\frac{da}{dN}=f\left(\sigma ,a,c\right)$$
where *a* is the length of the fatigue crack; *N* is the number of loading cycles; σ is the normal stress; *C* is a parameter related to the material.

Paris proposed the relationship between the “d*a*/d*N*” and the magnitude of the stress intensity factor “ΔK” based on an experiment, which was the famous Paris formula [28].
(20)$$\frac{da}{dN}=C{\left(\Delta K\right)}^{m}$$
(21)$$K=Y\sigma {\left(a\right)}^{1/2}\;$$
where *a* is the length of the fatigue crack; *N* is the number of loading cycles; ΔK is the magnitude of the stress intensity factor of the cyclic load; $\Delta K=K_{\max}-K_{\min}$; *C* and *m* are parameters related to the materials, load conditions, and environmental factors; *Y* is the geometric or shape factor that reflects the geometric shape characteristics of the component, dimensionless; σ is the tensile stress, MPa; *A* is the length of the crack, m. 

According to the Paris formula, it can be seen that the “ΔK” is the main control parameter for crack growth, and the ratio of “d*a*/d*N*” will increase as “ΔK” increases. It is worth noting that the “ΔK” is also related to the crack length “*a*” according to its calculation in Formula (21). Then, both “d*a*/d*N*” and “ΔK” in the Paris formula are dependent variables related to the crack length “*a*”. However, the crack length “*a*” and the number of loading cycles “*N*” cannot conform to a well linear relationship in the fatigue crack growth stage according to the previous research. Therefore, it is incompatible with fitting the Paris formula directly based on the data of “d*a*/d*N*” and the magnitude of the stress intensity factor “ΔK”. In this paper, the Paris formula is fitted in two steps as follows: (1) The exponential relationships are fitted according to the *N-a* data, then the da/dN and its values are obtained by differentiating *a* to *N*. (2) The linear relationships are fitted according to lgΔk − lg(d*a*/d*N*), and then the values of parameter *C* and parameter *m* are obtained from the relationships. The fitting results are shown in Table 9.

Table 9 shows the fitting results based on the Paris formula under different curing times, and different models are applicable for different stress ratios. It can be seen that the values of parameters *C* and *m* in the formulas obtained from Table 9 both increase with the increase of the stress ratio at the 14-day curing. It explains that the crack length “*a*” is of great importance to the crack growth rate under high-stress levels at an early time. However, the 28-day and long-term curing do not reflect this rule. It is indicated that the fatigue cracking growth of the CRME is reflected by the cooperation effect of parameters *C* and *m* under a certain test condition.

### 3.6. Microscopic Morphology of Fracture Point 

The microstructure characteristics of the fracture surface under the SCB fatigue tests are analyzed by SEM with the backscattering method. The micro-topography of the fracture point of CRME in the fracture interface under different curing times is presented in Figure 11. 

As seen in Figure 11, the hydration products of the cement and asphalt membrane formed by the demulsification of the asphalt emulsion in the fracture interface can be clearly observed. However, the degree of the cement hydration and demulsification of the asphalt emulsion is different under different curing times. The asphalt membrane in Figure 11a is relatively dispersed and appears as smaller flaky particles. Compared with the microstructure characteristics of CRME at the 14-day curing in Figure 11a,b, it can be seen that the hydration products are significantly increased and almost coated by the asphalt membrane after the 28-day curing. The hydration products are composed of the asphalt membrane to form a denser spatial structure, which may provide a higher adhesive strength between the asphalt mortar and aggregate. Therefore, the fatigue cracking resistance of CRME is significantly increased. Moreover, the interface composed of hydration products and the asphalt membrane in Figure 11e,f is more uniform and has been further developed by accelerated curing. 

Based on the analysis of microstructure characteristics in the fracture surface of CRME under different curing times, it is concluded that the degree of cement hydration and demulsification of asphalt emulsion in the CRME is gradually increased and a homogeneous and dense microstructure is formed with the increase of the curing time. It is also indicated that the fatigue cracking resistance is gradually improved with the increase of the curing time. 

## 4. Conclusions

Fatigue cracking behavior and its evolution model of CRME under different curing times with different stress ratios are studied in this paper. The fatigue cracking model is discussed by means of the combination of damage mechanics and fracture mechanics. Based on the results above, the main conclusions are as follows:
(1)The fracture toughness $K_{IC}$ is used to characterize the crack resistance of CRME. CRME has a poor cracking resistance in the early time, and it gradually increases with the increase of the curing time.(2)The fatigue equation of cold recycled mixtures is obtained by fitting the regression equation of $\mathrm{lg}N_{f}-\sigma$, and the value of coefficient k is used to reflect the fatigue cracking resistance. The value of *k* gradually increases with the increase of the curing time, which indicates that the fatigue performance is gradually improved with the increase of the curing time.(3)The ratio of the “initial cracking point/fatigue life” under the same curing time increases as the stress ratio increases. In addition, under the same stress ratio, the ratio shows a decreasing trend with the increase of the curing time. The initial fatigue cracking on the surface of the sample appears at around 60% of the fatigue life.(4)The fatigue damage model based on damage mechanics and the fatigue loading equation of the CRME is established. The damage variable is only between 0.06–0.17 when the initial fatigue cracks begin to start during the loading process.(5)The Paris formula based on fracture mechanics reflects the relationships between the crack growth rate and stress intensity factor. The crack length “*a*” has a greater impact on the crack growth rate under high-stress levels at the early stage. In contrast, crack growth at the 28-day and long-term curing is controlled by the cooperation effect of parameters *C* and *m*.(6)The degree of cement hydration and demulsification of asphalt emulsion in CRME is gradually increased, and a denser spatial structure is gradually formed during the curing times, which provides higher fatigue cracking resistance of CRME.

## Figures and Tables

**Figure 1 materials-15-04476-f001:**
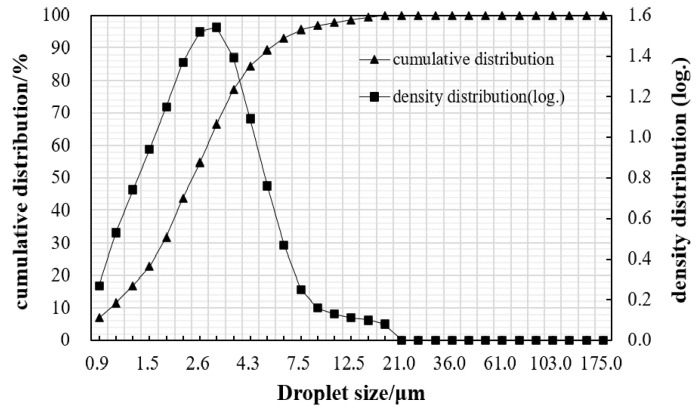
The droplet size distribution of asphalt emulsion.

**Figure 2 materials-15-04476-f002:**
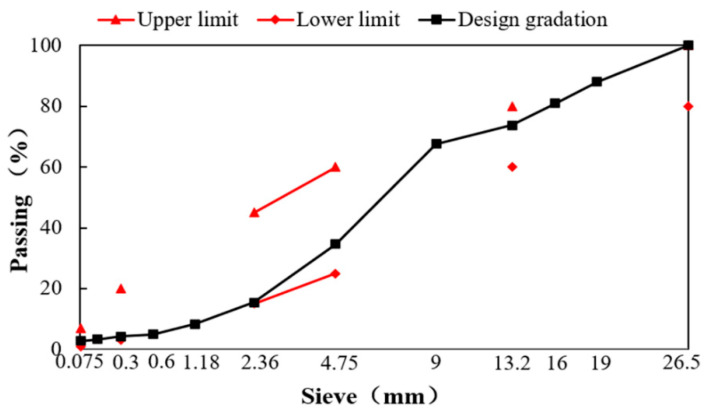
The gradation of CRME.

**Figure 3 materials-15-04476-f003:**
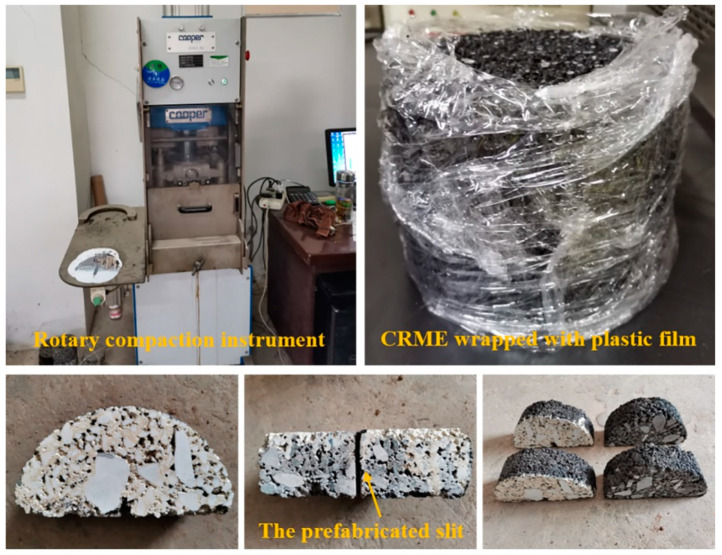
The preparation process of CRME samples.

**Figure 4 materials-15-04476-f004:**
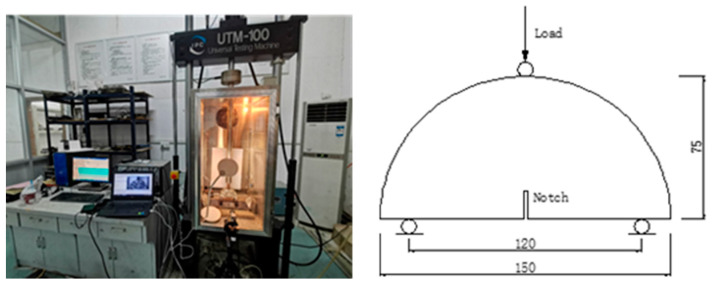
The SCB test loading system and the force mode.

**Figure 5 materials-15-04476-f005:**
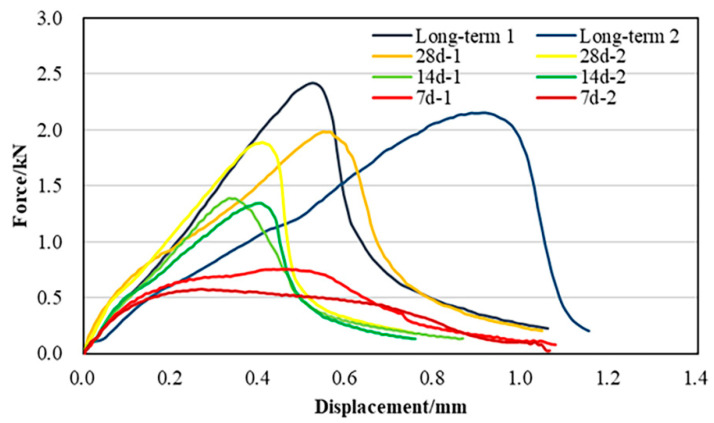
The load-displacement curve of CRME at different curing times.

**Figure 6 materials-15-04476-f006:**
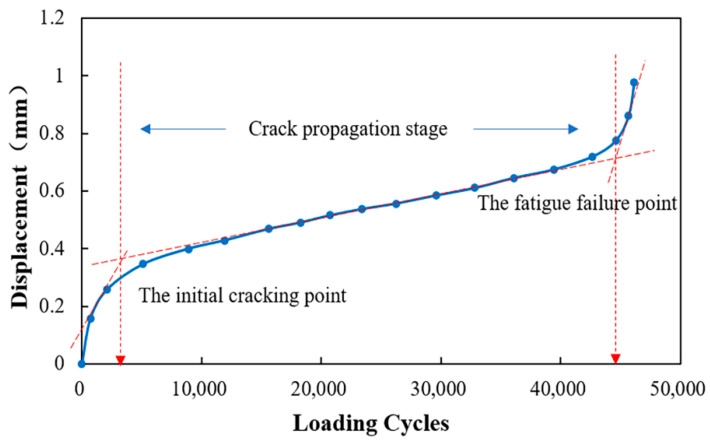
The typical fatigue loading curve of asphalt mixtures.

**Figure 7 materials-15-04476-f007:**
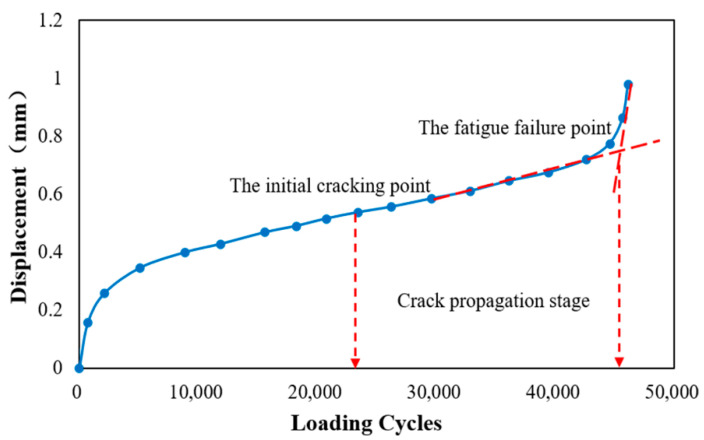
The position of the initial cracking point and fatigue failure point in the loading curve.

**Figure 8 materials-15-04476-f008:**
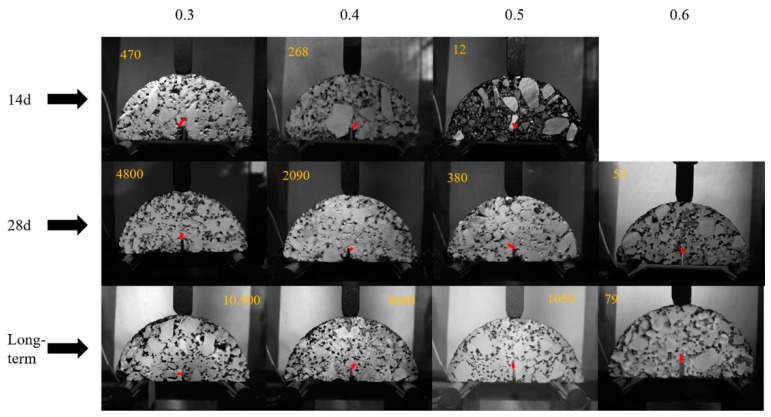
The position of the initial cracking point with different stress ration.

**Figure 9 materials-15-04476-f009:**
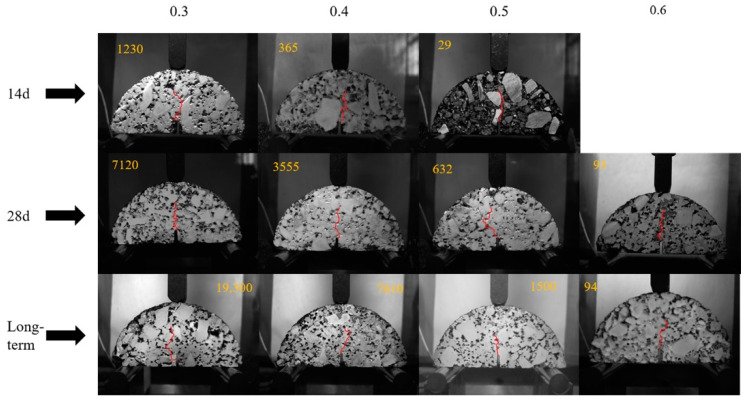
The fatigue failure point of the samples with different stress ration.

**Figure 10 materials-15-04476-f010:**
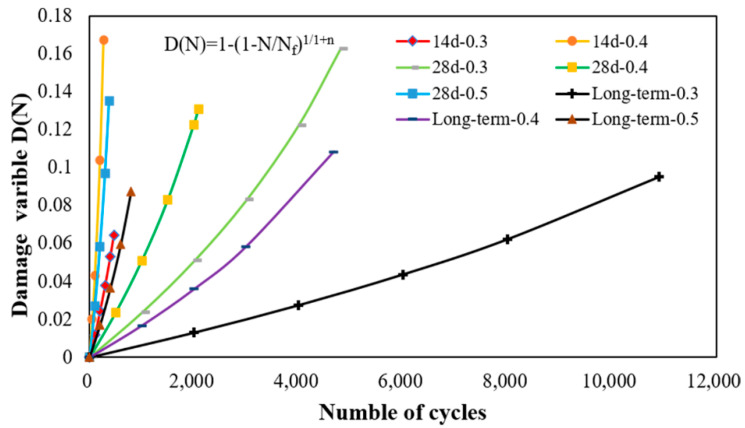
Damage evolution equations under different curing times with different stress ratios.

**Figure 11 materials-15-04476-f011:**
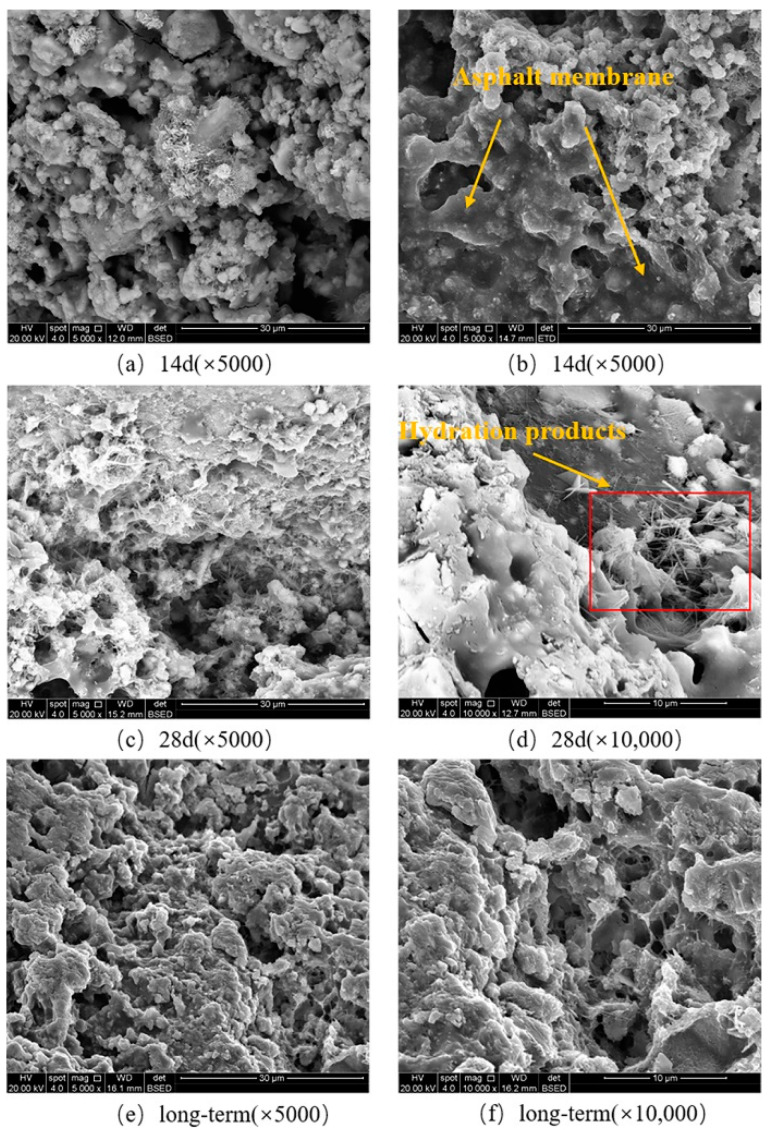
The micro-topography of the fracture point of CRME with different curing times.

**Table 1 materials-15-04476-t001:** RAP aggregates gradation of CRME.

Sieve (mm)	26.5	19	16	13.2	9	4.75	2.36	1.18	0.6	0.3	0.15	0.075
RAP 10–20	100	90.9	58.6	25.7	7.0	0.2	0.2	0.2	0.2	0.2	0.2	0.2
RAP 5–10	100	100	100	100	94.7	7.2	0.1	0.1	0.1	0.1	0.1	0.1
RAP 0–5	100	100	100	100	100	99.8	37.4	15.3	6.6	4.8	2.4	1.8

**Table 2 materials-15-04476-t002:** The properties of asphalt emulsion.

Property	Unit	Value	Technical Requirements
Demulsification speed	-	Slow-breaking	Slow-breaking
Particle charge	-	Cation (+)	Cation (+)
Residue on sieve (1.18 mm)	%	0.02	≤0.1
Enguera Viscosity E_25_	-	5.9	2–30
Residue by distillation	%	62.5	≥60
Penetration of residue, 25 °C	0.1 mm	58.2	50–130
Ductility of residue, 15 °C	cm	69.5	≥40
Softening point of residue	°C	51.3	≥46
Adhesion with coarse aggregate, coating area	-	≥2/3	≥2/3
Storage stability at 1 day	%	0.1	≤1
Storage stability at 5 days	%	0.9	≤5

**Table 3 materials-15-04476-t003:** The composition of aggregate.

New Aggregate 19–26.5	RAP 10–20	RAP 5–10	RAP 0–5	New Aggregates 0–5	Filler
10%	22%	36%	24%	6%	2%

**Table 4 materials-15-04476-t004:** The calculation of fracture parameters.

Curing Time (d)	Groups	F_max_ (kN)	$\sigma_{\max}\;(\mathrm{MPa})$	$K_{IC}$ (MPa·mm^1/2^)	G_f_ (J/m^2^)	FI	CRI (m^−1^)
7	1	0.75	0.48	17.41	198.96	0.94	0.26
	2	0.58	0.37	13.46	167.50	1.05	0.29
14	1	1.39	0.89	32.26	210.25	0.25	0.15
	2	1.35	0.86	31.33	196.71	0.11	0.15
28	1	1.98	1.27	45.95	412.75	0.32	0.21
	2	1.89	1.21	43.86	264.17	0.07	0.14
Long term	1	2.42	1.55	56.16	440.46	0.20	0.18
	2	2.15	1.38	49.90	579.79	0.28	0.27

**Table 5 materials-15-04476-t005:** The fatigue life of CRME with different curing times under different stress ratios.

Curing Time (d)	Stress Ratios	Stress (MPa)	Fatigue Life
14	0.2	0.17	15,630
	0.3	0.26	1230
	0.4	0.35	365
	0.5	0.44	29
28	0.2	0.25	45,300
	0.3	0.37	7120
	0.4	0.50	3555
	0.5	0.62	632
	0.6	0.74	93
Long-term	0.3	0.44	19,300
	0.4	0.58	7610
	0.5	0.73	1500
	0.6	0.88	94

**Table 6 materials-15-04476-t006:** Fatigue equation of CRME under different curing times.

Curing Time	Fatigue Equation	*n*	*k*	R^2^
14-day	$\mathrm{lg}N_{f}=-6.2659\mathrm{lg}\sigma -0.5696$	6.27	0.27	0.99
28-day	$\mathrm{lg}N_{f}=-5.3473\mathrm{lg}\sigma +1.5725$	5.35	37.37	0.99
Long-term	$\mathrm{lg}N_{f}=-7.3323\mathrm{lg}\sigma +1.8972$	7.33	78.92	0.92

**Table 7 materials-15-04476-t007:** The characteristic of the initial cracking point and failure point of CRME.

Curing Time (d)	Stress Ratio	Initial Cracking Point	Fatigue Life	Initial Cracking Point/Fatigue Life
14	0.3	470	1230	0.38
	0.4	268	365	0.73
	0.5	12	29	0.41
28	0.2	23,500	45,300	0.52
	0.3	4800	7120	0.67
	0.4	2090	3555	0.59
	0.5	380	632	0.60
	0.6	53	93	0.57
Long-term	0.3	10,900	19,300	0.56
	0.4	4680	7610	0.62
	0.5	1050	1500	0.70
	0.6	79	94	0.84

**Table 8 materials-15-04476-t008:** The fitting results of the damage variable.

Curing Time (d)	Stress Ratio	N/N_f_	*n*	D(N)
14	0.3	0.38	6.27	0.06
	0.4	0.73		0.17
28	0.3	0.67		0.16
	0.4	0.59	5.35	0.13
	0.5	0.60		0.13
Long-term	0.3	0.56		0.09
	0.4	0.62	7.33	0.11
	0.5	0.70		0.13

**Table 9 materials-15-04476-t009:** The fitting results of the Paris formula.

Curing Time (d)	Stress Ratio	Paris Formula
14	0.3	$\frac{da}{dN}=3.67\times 10^{-4}{\left(\Delta K\right)}^{1.6687}$
	0.4	$\frac{da}{dN}=1.14\times 10^{-3}{\left(\Delta K\right)}^{1.8314}$
	0.5	$\frac{da}{dN}=2.64\times 10^{-3}{\left(\Delta K\right)}^{1.9336}$
28	0.2	$\frac{da}{dN}=3.8\times 10^{-7}{\left(\Delta K\right)}^{2.4399}$
	0.3	$\frac{da}{dN}=1.45\times 10^{-5}{\left(\Delta K\right)}^{2.14}$
	0.4	$\frac{da}{dN}=7.65\times 10^{-5}{\left(\Delta K\right)}^{1.6866}$
	0.5	$\frac{da}{dN}=1.48\times 10^{-4}{\left(\Delta K\right)}^{1.8298}$
	0.6	$\frac{da}{dN}=1.71\times 10^{-3}{\left(\Delta K\right)}^{1.5289}$
Long term	0.3	$\frac{da}{dN}=4.69\times 10^{-6}{\left(\Delta K\right)}^{2.2775}$
	0.4	$\frac{da}{dN}=5.92\times 10^{-7}{\left(\Delta K\right)}^{2.8067}$
	0.5	$\frac{da}{dN}=3.50\times 10^{-7}{\left(\Delta K\right)}^{3.2597}$
	0.6	$\frac{da}{dN}=4.18\times 10^{-5}{\left(\Delta K\right)}^{2.7399}$

## Data Availability

Not applicable.

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
