# Peer review of "Fatigue Cracking Evolution and Model of Cold Recycled Asphalt Mixtures during Different Curing Times"

_materials, 2022, doi:10.3390/ma15134476_

Round 1

Reviewer 1 Report

The manuscript is well structured. One of the important advantages of this paper is the study of fatigue in cold asphalt mixtures. Here are some comments to increase improvement:

1-      In section 2-1-1 what is the JTG…. Standard?

2-      In section 2.1.3, has portland cement been used as a filler in this research? Please explain clearly.

3-      What is the asphalt mix design in this study? Percent of emolusion,…void of ratio?

4-      In section of 2.3.2, Describe the capabilities of the CMOS camera further.

Reviewer 2 Report

This paper studies the fatigue cracking evolution of cold recycled asphalt mixtures that have been prepared with asphalt emulsion (CRME). In addition different curing times have been analysed.

The experimental work,  is well conceived and conducted, data interpretation is reliable, referencing is appropriate.

Although interesting results are obtained, the manuscript raises several questions. Here are some comments on specific points:

·         Extra details of  Asphalt emulsion must be reported: continuous and dispersed phase concentration, emulsified used, preparation method, droplet size distribution etc

·         As cracking is a key point of the research, low temperature properties of the asphalt residue  must be reported: Fraass point, glass transition temperatures, BBR etc

·         Temperature of Curing time test must be reported

·         As fatigue tests have been performed at 15ºC, bitumen properties at this temperature should be  described. In addition, bitumen fatigue tests is also interesting

·         It not clear from the results what is the contribution of mineral aggregates and bitumen to the global behaviour of the mixture  

·         Fig 10. The so called asphalt membrane is not clearly visualised

·          
